# Socioeconomic and demographic patterning of family uptake of a paediatric electronic patient portal innovation

**Ameenat Lola Solebo**[1,2,3,4]*, **Lisanne Horvat-Gitsels**[1,3,5], **Christine Twomey**[2], **Siegfried Karl Wagner**[6,7], **Jugnoo S. Rahi**[1,2,3,4,6,7,8]

**1** Population, Policy and Practice Research and Teaching Department, Great Ormond Street Institute of Child Health, University College London, London, United Kingdom, **2** Great Ormond Street Hospital for Children NHS Foundation Trust, London, United Kingdom, **3** Ulverscroft Vision Research Group, Great Ormond Street Institute of Child Health, University College London, London, United Kingdom, **4** National Institute for Health and Care Research Great Ormond Street Biomedical Research Centre, London, United Kingdom, **5** Moody's RMS, London, United Kingdom, **6** Moorfields Eye Hospital, London, United Kingdom, **7** Institute of Ophthalmology, University College London, London, United Kingdom, **8** NIHR Moorfields Biomedical Research Centre London, United Kingdom

* a.solebo@ucl.ac.uk

**Data Availability Statement:** Individual level data are not being made available for this study involving human research participant data. Due to the potentially identifiable nature of the dataset and

## Abstract

Patient portals allowing access to electronic health care records and services can inform and empower but may widen existing sociodemographic inequities. We aimed to describe associations between activation of a paediatric patient portal and patient race/ethnicity, socioeconomic status and markers of previous engagement with health care. A retrospective single site cross-sectional study was undertaken to examine patient portal adoption amongst families of children receiving care for chronic or complex disorders within the United Kingdom. Descriptive and multivariable regression analysis was undertaken to describe associations between predictors (Race/Ethnicity, age, socio-economic deprivation status based on family residence, and previous non-attendance to outpatient consultations) and outcome. A sample of 3687 children, representative of the diverse 'real world' patient population, was identified. Of these 37% (1364) were from a White British background, 71% (2631) had English as the primary family spoken language (PSL), 14% (532) lived in areas of high deprivation, and 17% (643) had high (>33%) rates of non-attendance. The families of 73% (2682) had activated the portal. In adjusted analyses, English as a PSL (adjusted odds ratio [aOR] 1.58, 95% confidence interval 1.29–1.95) and multi-morbidity (aOR 1.26, 1.22–1.30) was positively associated with portal activation, whilst families from British Black African backgrounds (aOR 0.68, 0.50–0.93), and those with high rates of non-attendance (aOR 0.48, 0.40–0.58) were less likely to use the portal. Family race/ethnicity and previous low engagement with health care services are potentially key drivers of widening inequity in access to health care following the implementation of patient portals, a digital health innovation intended to inform and empower. Health care providers should be aware that innovative human-driven engagement approaches, targeted towards previously underserved communities, are needed to ensure equitable access to high quality patient-centred care.

as participants have not given the necessary consent, the authors are prevented from publicly sharing individual level data under the institutional approvals for this study from Great Ormond Street Hospital NHS Foundation Trust. New ethics approval would be needed to either access individual level data without participant consent, or to re-approach participants in order to seek consent for data sharing. For more information, contact the Great Ormond Street Hospital Research Management and Governance Team (RM&G Team) research.administration@gosh.nhs.uk.

**Funding:** This work was supported by an NIHR Clinician Scientist award (CS-2018-18-ST2-005, to ALS); and an MRC Clinical Research Training Fellowship (MR/TR000953/1, to SW). The funders had no role in study design, data collection and analysis, decision to publish, or preparation of the manuscript.

**Competing interests:** The authors have declared that no competing interests exist.

## Author summary

From a retrospective cross-sectional study of 3687 children with complex health disorders within a specialist paediatric care centre, the families of 73% had adopted an electronic patient portal within 2.5 years of portal launch. Family ethnic backgrounds and previous poor engagement with health care services were independently associated with lower odds of family adoption. There was evidence of a potential differential impact of socioeconomic deprivation and spoken language across different ethnic groups. We suggest that equitable uptake of digital health services by children's families requires health care providers to implement engagement approaches developed in partnership with underserved communities. However, those underserved communities should also have access to alternative patient centred communication pathways to ensure true inclusion in health care provision. Care providers must be particularly careful to offer these alternative pathways to families who have struggled to interact with healthcare in the past.

## Introduction

An individual's right to a computable version of their medical records is a central principle in digital health policy [1]. These computable versions can, in some settings, be accessed via a 'patient portal' which allows individuals to view, download, contribute to and share their electronic health records, or use those records to self-manage their health conditions and broader wellbeing [2].

The positive impact on patient outcomes and experience seen following digital health innovations can be weakest for those individuals in their country's more vulnerable socioeconomic strata [3–5]. There is a real risk that the increasing use of patient portals widens existing socioeconomic and demographic inequalities in health outcomes or access to healthcare [6,7].

An under-represented patient population vulnerable to this potential digital divide are children. As with adults [8], limited engagement with primary care patient portals has been observed in the paediatric population, with low-income families and those from ethnic minority groups demonstrating reduced engagement [9,10]. However, there has been little investigation into children with complex medical needs and multimorbidity. Such features are common among those children affected with, or at risk of, visual impairment. Childhood visual impairment (VI), which is a health outcome as well as a medical condition, has an annual incidence of 1 per 1000 [11]. VI in childhood is a strong marker of broader child health, with close correlation with child mortality, and with 72% of children with VI also having other non-ophthalmic disorders or impairments [11,12]. It also shares with broader child health a strong patterning with socioeconomic family status. In the UK, socioeconomic deprivation results in increased risk of childhood blindness [11], reduced access to innovations in care despite the universal healthcare system [13], and worse outcomes following interventions [14].

There is a lack of evidence on the patterns of adoption for patient portals and similar health technologies within this population. We used the opportunity created by the COVID-19 pandemic, which triggered rapid pivoting to use of patient portals to ensure sustained communications with patients, to test our hypothesis that differential adoption, by socioeconomic status and demographic characteristics, of a patient portal service exists amongst a group of families receiving specialist paediatric care.

## Results

We identified 3687 eligible patients, of whom 48% were female (Table 1).

The median age of the patient cohort was 8 years (interquartile range, IQR 4 to 12, range 0 to 18 years). Of the 3687, 191 young people (5%) were aged 16 years or older. There were a diverse range of eye conditions, including inflammatory ocular disease (796, 22%), cataract (342, 9%), and glaucoma (132, 4%). Non-ocular disorders or impairments had been diagnosed in the majority of children, with the number of additional disorders ranging from 1 to 21 (median 3).

**Table 1. Patient characteristics.**

| Characteristic | All patients (n = 3687) |
|---|---|
| Sex—no. (%) | |
| Female | 1769 (48) |
| Male | 1919 (52) |
| Missing | 0 |
| Age—no. (%) | |
| Under 2 (<2yrs) | 331 (9) |
| Pre-school (2–4) | 640 (17) |
| Early childhood (5–10) | 1423 (39) |
| Late childhood (11yrs and older) | 1293 (35) |
| Missing | 0 |
| [a]Ethnicity–no. (%) | |
| White British | 1364 (37) |
| Other White background | 392 (11) |
| Asian Bangladeshi | 128 (3) |
| Asian Indian | 147 (4) |
| Asian Pakistani | 251 (7) |
| Other Asian background | 189 (5) |
| Black African | 279 (7) |
| Other Black background | 177 (5) |
| Mixed ethnicity | 191 (5) |
| Other ethnic background | 338 (9) |
| Prefer not to say / Not provided | 231 (7) |
| Primary language spoken–no. (%) | |
| English | 2631 (71) |
| Other language | 826 (23) |
| Prefer not to say / Not provided | 230 (6) |
| Family residence index of multiple deprivation (IMD) quintile–no. (%) | |
| Most deprived (1st quintile) | 623 (17) |
| 2nd quintile | 1072 (29) |
| 3rd quintile | 768 (21) |
| 4th quintile | 600 (16) |
| Least deprived | 532 (14) |
| Unknown (primary residence outside the UK) | 98 (3) |
| High non-attendance rate | 643 (17) |
| Additional non-ophthalmic disorders or impairments | 2896 (79) |

[a]Comparative population level frequency data on ethnicity of UK children: White British 73%; Asian British 12%; Black British 5%

### Cohort sociodemographic characteristics

There was a wide range of ethnicities and languages spoken (Table 1, S1 and S2 Tables): 1364 children (37%) were of White British ethnicity, and 2631 children (71%) were from families with English as the primary spoken language. Non-attendance to hospital appointments over the preceding year ranged from 0 to 100% (median 0%, IQR 0–20%) and 643 children (17%) had not been brought to more than 33% of their preceding outpatient appointments. A higher percentage of children with English as a primary language had a White British rather than other ethnic background (95% versus 57% from a non-white background, Pearson $\chi^2$ = 600.1, $p<0.001$). A higher percentage of children whose primary language was not English lived in a deprived area (20% versus 16% of children with English as a primary language, Pearson $\chi^2$ = 12.6, $p<0.001$). Deprivation was strongly associated with ethnicity: whilst 11% of families from White British and 9% from Asian Indian backgrounds lived in areas of deprivation (156/1244, and 14/147 respectively), this proportion was 19% in families from Asian Pakistani backgrounds (48/251), 34% in Asian Bangladeshi background families (43/128), 33% for Black African background families (93/279), and 30% (66/225) for families from other Black background (Pearson $\chi^2$ = 141.7, $p<0.001$). There was no statistically significant association between deprivation and high non-attendance rate (Pearson $\chi^2$ = 2.8, $p = 0.10$) but non-attendance was less likely to be high amongst those families with a high number of additional non-ophthalmic disorders (unadjusted odds ratio 0.80, 95% CI 0.78 to 0.83, $p<0.001$).

### Outcomes

Most children (2682, 73%) had families who had activated the patient portal. The median time to activation from portal launch was 13 months (range from 0.5 days to 30 months). A total of 2399 families had used the portal over this period (comprising 65% of the whole cohort and 90% of those who had activated the portal).

The unadjusted and adjusted odds of patient activation of the portal for the variables of interest, as presented in Table 2, were estimated for a total of 3416 children in complete case analyses (i.e., at least one covariable was missing in 271/3687 cases).

Families who had a high non-attendance at outpatient clinics had half the odds (adjusted odds ratio [aOR] 0.48, 95% confidence interval [CI] 0.40–0.58) of activating the patient portal when compared to other families, with the association proving robust to adjustment for clinical need (the number of associated non-ophthalmic conditions). Higher odds of activating the portal were associated with having English as a primary language (1.58, 1.29–1.95). Families from an Asian Indian (but not other Asian) background were also more likely to have activated the portal than White British origin families (1.65, 1.05–2.68). However, families from Black African backgrounds had lower odds of activation (0.68, 0.50–0.93, respectively) with evidence of differential impact of language and deprivation across different family ethnic groups (Fig 1).

The associations between covariables and the secondary outcome of family use of the portal was estimated for the 2626 families who had activated the protocol (full dataset on covariables missing of 56/2682). On adjusted analysis, the absence of English as a primary family language, high non-attendance rate for previous consultations, and older age of child (11 years and older) were all independently associated with lower odds of using the portal (Table 3), whilst diagnoses of additional non-ophthalmic disorders were associated with portal use.

### Discussion

From this cross-sectional study based within a specialist paediatric centre serving a diverse patient population, we report differential adoption of an electronic patient portal by sociodemographic characteristics. In the 30 months following the launch of the portal, the families

**Table 2. Associations between characteristics of the patients and family uptake of the portal.**

|  | Odds Ratio (95% confidence interval), *p*-value | |
|---|---|---|
|  | **Unadjusted OR** | **Adjusted OR** |
| Age group |  |  |
| *Children aged under 2* | Reference | Reference |
| *Pre-school (2 to under 5yrs)* | 1.02 (0.77–1.34) | 0.90 (0.66–1.23), 0.53 |
| *Early childhood (5 to under 11yrs)* | 1.19 (0.92–1.52) | 1.03 (0.78–1.37), 0.83 |
| *Late childhood (11yrs and older)* | 0.93 (0.73–1.20) | 1.82 (1.34–2.49), <0.01 |
| Female Sex | 0.95 (0.84–1.10) | - |
| High non-attendance (>33% missed appointments) | 0.34 (0.29–0.40) | 0.48 (0.40–0.58), <0.001 |
| Ethnicity |  |  |
| *White British* | Reference | Reference |
| *Asian Indian* | 1.29 (0.82–2.00) | 1.65 (1.05–2.68), 0.03 |
| *Asian Pakistani* | 0.68 (0.51–0.92) | 0.65 (0.46–1.14), 0.30 |
| *Asian Bangladeshi* | 1.14 (0.73–1.79) | 1.39 (0.84–2.30), 0.19 |
| *Black African* | 0.59 (0.44–0.78) | 0.68 (0.50–0.93), 0.02 |
| *Black Other* | 0.73 (0.54–1.00) | 1.00 (0.94–1.14), 0.27 |
| *Ethnicity not provided* | 0.21 (0.16–0.27) | 0.55 (0.43–0.67), 0.03 |
| English as primary spoken language | 1.80 (1.60–2.20) | 1.58 (1.29–1.95)[¥], <0.001 |
| Residence in area of relative deprivation | 0.80 (0.60–0.90) | 0.83 (0.67–1.03), 0.09 |
| Additional non-ophthalmic disorders or impairments | 1.27 (1.23–1.30) | 1.26 (1.22–1.30), <0.001 |

[¥]Interaction terms with ethnicity, and with deprivation, not statistically significant

of the majority of children had accessed and used the application. Adoption was higher amongst families with English as a primary spoken language and for families of older children. Families from Black ethnic backgrounds were less likely to adopt the application, as were those who opted to withhold details of their ethnicity. The strongest association with uptake and subsequent use of the portal was with prior poor attendance to outpatient consultations, with

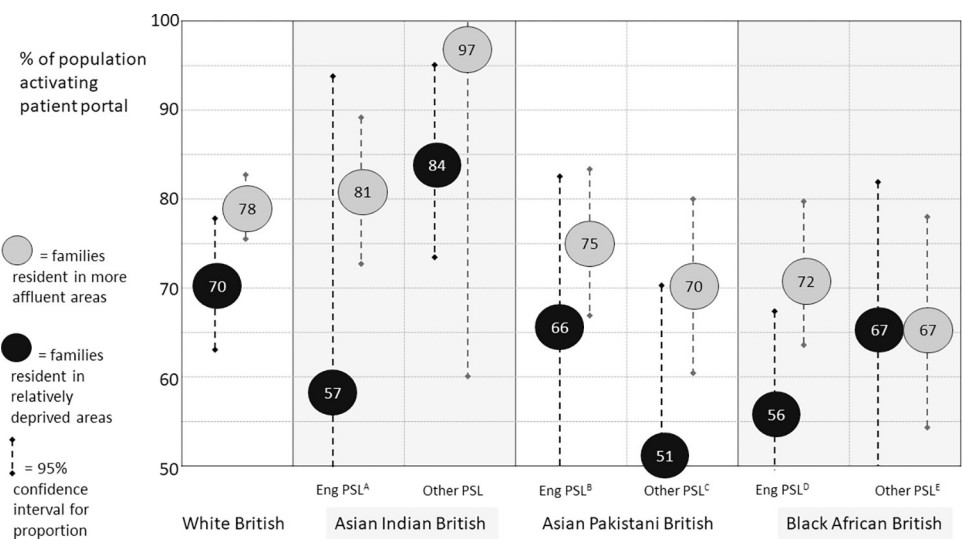

**Fig 1. Uptake of the patient portal by family socio-demographic characteristics.** *Eng PSL*: *Family with English as primary spoken language; Other PSL*: *Family with language other than English as primary spoken language.* A: 95% confidence interval, CI 20–94; B: 95% CI 48–83; C: 95% CI 25–70; D: 95% CI 43–69; E: 95% CI 51–82.

**Table 3. Associations between characteristics of the patients and family use of the portal given uptake.**

| | Odds Ratio (95% confidence interval), *p*-value | |
| --- | --- | --- |
| | **Unadjusted** | **Adjusted** |
| Age group | | |
| *Children aged under 2 (baseline)* | Reference | Reference |
| *Pre-school (2 to under 5yrs)* | 1.54 (0.82–2.88) | 0.92 (0.67–1.26), 0.61 |
| *Early childhood (5 to under 11yrs)* | 2.01 (1.15–3.52) | 1.11 (0.84–1.48), 0.47 |
| *Late childhood (11 and older)* | 0.56 (0.33–0.94) | 0.42 (0.30–0.59), <0.001 |
| Female sex | 1.09 (0.85–1.40) | - |
| High non-attendance (>33% missed appointments) | 0.56 (0.41–0.78) | 0.48 (0.39–0.58), <0.001 |
| Ethnicity | | |
| *White British* | Reference | Reference |
| *Asian Indian* | 1.38 (0.65–2.90) | 1.74 (1.14–2.67), 0.01 |
| *Asian Pakistani* | 1.05 (0.60–1.86) | 0.78 (0.57–1.07), 0.12 |
| *Asian Bangladeshi* | 0.57 (0.32–1.02) | 1.15 (0.74–1.79), 0.54 |
| *Black African* | 0.46 (0.30–0.70) | 0.61 (0.46–0.82), 0.001 |
| *Black Other* | 0.78 (0.57–1.06) | 0.96 (0.78–1.17), 0.66 |
| *Ethnicity not provided* | 0.62 (0.34–1.10) | 0.83 (0.60–1.15), 0.27 |
| English as primary spoken language | 1.65 (1.27–2.15) | 1.69 (1.41–2.05), <0.001 |
| Residence in area of relative deprivation | 0.87 (0.63–1.20) | 0.88 (0.72–1.08), 0.21 |
| Additional non-ophthalmic disorders or impairments | 1.22 (1.19–1.24) | 1.20 (1.17–1.23), <0.001 |

two times lower odds of adoption amongst families with high rates of non-attendance, exemplifying an amplification of inequalities in health care use. These factors were independent of socioeconomic status as measured using location of family residence, and independent of clinical need as measured using a quantification of multi-morbidity.

Our findings on the socioeconomic patterning of portal adoption are consistent with earlier reports from adult [8] and paediatric care settings [9,10]. These North American studies have reported lower adoption amongst patients from Africa American [8], Black [9] and Hispanic backgrounds [10], those who are uninsured [9,10] or who have other markers of socio-economic deprivation [8–10]. Through our study population, we are able to report similar patterning within a high income setting despite a universal healthcare system, and brought increased focus to the potential differential use across different ethnic minority families and those who have additional markers of dis-engagement (non-attendance, or withholding demographic data on ethnicity). Whilst previous primary care based studies were not able to find an independent association between multi-morbidity and paediatric care portal use [10], our 'exemplifier' population of children with complex care needs suggest that multi-morbidity is indeed a driver for portal adoption.

Across the globe, in diverse health care settings, socioeconomic and demographic factors such as income level and ethnicity have a marked influence on access to health services and subsequent health outcomes [3,15]. Access to digital health services is also dependent on the design of those services, and the degree to which these are centred on the life experience of the patients they serve. The life experiences of those developing and implementing patient applications may differ drastically from many of those accessing these services [5]. One in six adults resident in Organization for Economic Co-operation and Development (OECD) countries is unable to make the inferences necessary to match short digital or printed text to a piece of information (Level 1 literacy) [16,17]. Digital literacy (confidence in using the technology) is only one of the multiple factors at play when patients or families do not engage with health

technology. Digital literacy, in particular, has been identified as a determinant of an individual's physical and mental health and quality of life, and can be improved through education and training (e.g. through online courses, tutoring, and video-based training of children and adults) or through social support from 'digitally confident' peers or family members [18]. Digital access (access to a functional device or the internet) and digital assimilation (awareness and/or trust in the degree to which the technology can have a positive impact on outcomes for the individual patient or family) are also important factors [19]. Whilst individual care providing professionals may be unable to improve their patient's digital literacy or access, our findings suggest that professionals should be aware of these potential obstacles. We recommend that they build this awareness, and amend their approach accordingly. In a manner analogous to the use of a novel therapy found to be effective at a population level, but also known to have differential effectiveness in certain patient groups, health care professionals should 'prescribe' health technology innovations with full awareness of the 'relative contraindications' at play for their patient. Non-digital complex interventions and innovation are also needed: we recommend that care providers (at individual and organisational levels) should develop and deploy other ways of (re)connecting with families who have disengaged to the extent that children are not being brought to clinic appointments, either because of active disengagement with health care provision, or passive disengagement due to difficulties in accessing travel resources, childcare for siblings, or time off work. These children will otherwise be at increasing risk of being further 'left behind', with life-long, cumulative negative impact on their adult functioning and quality of life.

Digital assimilation, i.e. improving the patient's awareness of the potential positive impact of the digital health service, or increasing their trust in that service, is the area where health care providers can have the greatest positive impact on population adoption of digital health services. Lower levels of trust in the UK's National Health Service (NHS) are reported from individuals from ethnic minority backgrounds [20], with similar reports from other health care settings [21–23]. Whilst these underserved communities might believe in principle with using health data to improve health outcomes, their negative experiences of the healthcare system may limit their willingness to engage personally with system-level health innovations. This is likely to be particularly true for Black parents who have had to navigate a health system in which Black infants, and Black mothers, have worse outcomes, such as a mortality rate several times higher than the national average [24,25]. These differential mortality rates are likely to be accompanied by higher morbidity rates for Black children and mother.

To avoid widening inequity, we recommend that health services urgently implement targeted engagement and trust-building activities for those communities who have been given good reason to distrust an organisation in which there is longstanding inequity of outcome. These activities might include co-development of dissemination strategies to demonstrate the resultant benefit to care provision for the specific individual using the available frameworks for such activities [26], and transparency of any data usage and sharing for purposes other than direct clinical care.

The striking failure of digital patient portals to engage those families who struggled to bring their children to clinic appointments reflects the key importance of assessing and responding to the markers of non-engagement with health care services across different service types. A 'soft' marker of non-engagement may be reluctance to self-provide family ethnic background [27], which may again also be a marker of that family's lack of trust in care providers. The higher rates of non-engagement with this e-health intervention we report amongst those families who have not self-reported their ethnicity is similar to the association between higher non-attendance (across face to face and telemedicine services) and lack of ethnicity self-reporting reported by other UK groups [27].

Competing health and social needs may have driven some of the differential patterning of portal activation and portal use. Whilst families of older children were more likely to activate the portal, they were less likely to use the portal once activated. It may be that families of younger children have more competing demands on their time, with a resultant lower likelihood of activating the portal, but once activated, the greater clinical needs of the younger children [28,29] result in greater use of the portal. Again, awareness of the possible competing needs of the families accessing these digital health services is necessary to ensure the development of approaches in communicating the potential benefit of these services to those families in order to support digital assimilation.

## Limitations

This work is limited by its single centre setting, as such work may result in findings which are not generalizable to other health care settings. However, as an early adopter (within the UK) of the increasingly used patient portal applications [4], this setting is uniquely well placed to report on key patterns of health technology use for UK families. Moreover, as a tertiary and quaternary healthcare centre providing otherwise unavailable specialist care to a national population, it is the ideal setting for studying the uptake of digital interventions by families of children with disorders sufficiently rare and complex as to require co-ordinated care across medical specialities and providers. Additionally, as care is provided to a national rather than regional population, a diverse patient cohort has been achieved, with a resultant representation of those groups who are often under-represented in work on electronic health applications. However, it should be stressed that the patterning reported here around ethnicity and use of the portal (i.e., different directions of association for those from Asian Indian versus Asian Pakistani or Black African backgrounds) does not obviate the likelihood of significant heterogeneity of use within these groups. Whilst this has been addressed in part through the use of covariables on language and residence-based (area-level) socioeconomic status, we are missing more granular (individual-level) detail such as parental income and education. We also lack a metric of the family's self-perceived need of, or benefit from, the use of the portal. It may be that particular families had children with unstable or severe disorders, or a high degree of patient anxiety and care-seeking behaviour, affecting their likelihood of adopting the portal. Conversely, it is possible that data on family PSL and ethnicity were not missing at random, and therefore reflected another aspect of family disengagement with health care services. Nevertheless, our study findings of differential adoption and lower adoption amongst underserved groups remain robust to these limitations, highlighting a service gap which needs to be addressed.

An additional element of the setting of this work is the time during which it occurred. The study period includes March 2020 to August 2021, during which time the United Kingdom was under 'lockdown' conditions to control viral transmission during the global pandemic. Patient portal services for access to EHRs were likely to have played a vital role during the pandemic by preventing disengagement with treatment or health care providers and supporting care coordination and telemedicine care delivery [30,31]. However, if the pandemic did result in higher overall rates of adoption and use of the patient portal, the differential adoption by different groups remains notable.

## Conclusion

In summary, care providers, from individual professionals to health care organisations, must remain aware of digital literacy and digital access constraints amongst the patients and families they serve, and work to support digital assimilation. Future national and global risks to care

delivery are expected, and health care providers must ensure that those digital interventions implemented to address these risks do not lead to a widening of health inequities. We also recommend that providers should be aware of the valid concerns of underserved patient populations around trust and transparency. 'Non-digital' or human-based innovations, such as patient peer support networks and patient liaison teams, must continue to play a role in empowering and informing those in need, and we recommend the ongoing development and implementation of these services. These patient and community centred approaches are also likely to be effective in addressing the 'paradox' of health innovations holding transformational potential for care delivery for those underserved population groups who are most likely to be excluded from the digital world [18].

## Materials and methods

The institutional approvals necessary to undertake this project (defined as a service evaluation study by the Institutional Clinical Quality Project board) were obtained, and this work adheres to the Helsinki Declaration recommendations.

### Study design

This retrospective, observational cross-sectional study included families of patients aged under 18 years managed by the ophthalmology specialities at an international centre of clinical excellence. As such, the hospital provides tertiary and quaternary level care to children with complex and or rare diseases. These children have been referred for specialised care by paediatric ophthalmologists elsewhere.

The hospital-developed patient portal, "MyGOSH", (a child- and UK-centred adaptation of the "MyChart" application) is integrated with the hospital EHR (Epic, Epic Systems Corporation), allowing parents and young people to access a subset of their clinical records, manage personal information, and send direct messages and images (e.g., reporting clinical change, querying prescriptions) to managing clinicians or administrative teams. MyGOSH launched in March 2019 [32], with invitations to adopt the portal sent to parents postally alongside physical displays of information throughout hospital grounds and informative videos on the hospital internet pages.

### Inclusion/Exclusion criteria

Paediatric ophthalmology was chosen as an 'exemplar' field, as the patient population is typified by the need for sustained communication co-ordinated across specialties and specialist care centres. The complex and rare childhood eye disorders managed in this care centre are managed nationally by only a small number of specialist centres [33], limiting the pool of informed health professionals available for consultation with families. These eye disorders are typically chronic, impactful and associated with other impairments and multi-system disorders [11,28], requiring co-ordination of care across multiple disciplines [28]. Primary care health professionals report a lack of confidence in managing even common eye disorders in these children [34,35], with affected families relying heavily on their specialist ophthalmology care team. There are socioeconomic and demographic inequalities in disease risk [11], access to novel therapies [13], and treatment outcomes in paediatric ophthalmology [14]. Consequently this population has the most to gain from innovations aimed at improving health experiences and outcomes.

The age of 18 years was used as a threshold as, within the UK, in line with the United Nations Convention on the Rights of the Child (CRC), a child is defined as anyone who has not yet reached their 18th birthday [36]. All families who received ongoing care from the

ophthalmology team (i.e., those who had scheduled outpatient consultations) from the time of the launch of the patient portal (March 2019) up until the end of the 30 months study period (September 2021) were included in this study. Families who had not received care (defined as inpatient or outpatient attendance) during the 6 months prior to the patient portal launch were excluded.

## Data collection

Data were collected from the electronic health record system, using the Epic SlicerDicer$^{TM}$ tool to develop the extraction report. The data extracted comprised the activation status of the MyGOSH patient application (i.e., whether or not the parent or guardian of the child had activated the portal to enable their use of the service; the study's primary outcome), deeper engagement with the patient portal (defined as the generation or submission of data by the patient, or a request for services; the study's secondary outcome), time to family activation from launch of the portal (in months), age (categorized using school age thresholds into children aged under 2 years, pre-school {2 to under 5 years}, early childhood or primary school age {5 to under 11 years}, late childhood or secondar school age{11 years and older}), sex at birth and race/ethnicity of patient (as defined by parents and as recorded at patient registration), postcode (zip code) of family residence, primary language spoken by in family home, and attendance rate at booked outpatient clinic appointments over the preceding year (categorized using an *a priori* determined threshold of 33% non-attendance to differentiate between high and low rates of non-attendance) [37]. The number of additional non-ophthalmic diagnoses or impairments was also quantified as a potential proxy marker of clinical need.

Socioeconomic status (SES) was derived using postcode conversion to an Indices of Multiple Deprivation (IMD) 2019 ranking. The IMD, the official measure of relative deprivation in England, capture relative levels of deprivation in 32,844 small areas in England across seven domains comprising income, employment, general health/disability, education, crime, barriers to housing and services, and living environment [38]. IMD ranking was then organized into deciles, and binarized using the lowest whole population based quintile: children resident in those areas were categorised as living in relative deprivation. Race/ethnicity was analysed using the original variable, and also binarized using the largest single group, thus into White British versus all other minority ethnicity groups, in order to examine patterns both by race/ethnic category (grouped using the United Kingdon Office of National Statistics, ONS, categories) and by whether or not the individual's family self-reported membership of an ethnic minority group. Primary spoken language was also binarized into English versus other language.

## Statistical analyses

We used descriptive statistics (frequency and proportion) to characterize the cohort. Missing data were investigated to understand potential bias in the cohort by testing their association with the other variables ($\chi2$). Logistic regression models of portal activation status (uptake, primary outcome) and deeper engagement (use, secondary outcome) were fitted to assess associations with sociodemographic factors, specifically race/ethnicity, language, IMD derived deprivation binarized rank, sex, age and clinic attendance. These models were fitted on complete cases. First, unadjusted associations were estimated. Multicollinearity of the independent variables was investigated using non-parametric tests ($\chi2$, Spearman's rank correlation coefficient) with alpha level set to 5% ($p < 0.05$ considered to be supportive of a correlation). Adjusted associations were then estimated, with the final adjusted models being derived using backward elimination to obtain the most parsimonious model possible without weakening

model fit. Where multicollinearity was present, the weakest association of the two relevant variables was removed from the adjusted model. We tested for interactions by incorporating two-factor interaction terms between included independent variables, with interactions retained if found to be statistically significant using the Wald test. All analyses were undertaken in Stata (Stata, version. 17.0; StataCorp).

## Supporting information

**S1 Table. Characteristics of included children by patient portal activation status.**
(DOCX)

**S2 Table. Primary spoken languages for families of children within study population.**
(DOCX)

## Acknowledgments

We are grateful to the patients and staff of the Ophthalmology and Rheumatology Departments at Great Ormond Street Hospital, with particular thanks to Mr Harry Petrushkin (Consultant Ophthalmologist), Ms Reshma Pattani (Specialised Optometrist) and Ilaria Testi (Consultant Ophthalmologist).

## Author Contributions

**Conceptualization:** Ameenat Lola Solebo, Christine Twomey.

**Data curation:** Ameenat Lola Solebo, Siegfried Karl Wagner, Jugnoo S. Rahi.

**Formal analysis:** Ameenat Lola Solebo, Lisanne Horvat-Gitsels, Siegfried Karl Wagner.

**Funding acquisition:** Ameenat Lola Solebo.

**Investigation:** Ameenat Lola Solebo, Christine Twomey, Jugnoo S. Rahi.

**Methodology:** Ameenat Lola Solebo, Siegfried Karl Wagner, Jugnoo S. Rahi.

**Resources:** Ameenat Lola Solebo.

**Supervision:** Ameenat Lola Solebo.

**Validation:** Ameenat Lola Solebo.

**Visualization:** Ameenat Lola Solebo.

**Writing – original draft:** Ameenat Lola Solebo.

**Writing – review & editing:** Ameenat Lola Solebo, Lisanne Horvat-Gitsels, Christine Twomey, Siegfried Karl Wagner, Jugnoo S. Rahi.

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
