## [Decision Letter · Decision Letter 0]

26 May 2024

PDIG-D-24-00122

Socioeconomic and demographic patterning of family uptake of a paediatric electronic patient portal innovation

PLOS Digital Health

Dear Dr. Solebo,

Thank you for submitting your manuscript to PLOS Digital Health. After careful consideration, we feel that it has merit but does not fully meet PLOS Digital Health's publication criteria as it currently stands. Therefore, we invite you to submit a revised version of the manuscript that addresses the points raised during the review process.

Please submit your revised manuscript within 60 days Jul 25 2024 11:59PM. If you will need more time than this to complete your revisions, please reply to this message or contact the journal office at digitalhealth@plos.org. Please include the following items when submitting your revised manuscript:

We look forward to receiving your revised manuscript.

Kind regards,

Calvin Or, PhD

Section Editor

PLOS Digital Health

Journal Requirements:

Additional Editor Comments (if provided):

Reviewers' comments:

Reviewer's Responses to Questions

**Comments to the Author**

1. Does this manuscript meet PLOS Digital Health’s publication criteria? Is the manuscript technically sound, and do the data support the conclusions? The manuscript must describe methodologically and ethically rigorous research with conclusions that are appropriately drawn based on the data presented.

Reviewer #1: Partly

Reviewer #2: Yes

Reviewer #3: Yes

2. Has the statistical analysis been performed appropriately and rigorously?

Reviewer #1: I don't know

Reviewer #2: Yes

Reviewer #3: Yes

3. Have the authors made all data underlying the findings in their manuscript fully available (please refer to the Data Availability Statement at the start of the manuscript PDF file)?

Reviewer #1: Yes

Reviewer #2: No

Reviewer #3: Yes

4. Is the manuscript presented in an intelligible fashion and written in standard English?

Reviewer #1: Yes

Reviewer #2: Yes

Reviewer #3: Yes

5. Review Comments to the Author

Reviewer #1: Introduction

The introduction states there are inequalities already within childhood blindness. Could these existing inequalities affect the ability to examine these relationships within the population included in this study? This could be included in the discussion section – perhaps in relation to a larger discussion about generalisability.

It might be helpful to include information about the incidence/prevalence of childhood blindness in the UK in the introduction for further context.

Materials and methods

The study design was described as cross-sectional, but based on the details in the methods I wondered if it might be a cohort study? Could this be clarified?

When examining the attendance rate at appointments over the past year, I was interested in knowing more about the distribution of the number of booked appointments per year in the patient population. Would it be possible a patient may not have a booked appointment and therefore non-attendance is not relevant? Was this variable explored as a possible predictor of engagement with the patient portal as it may indicate greater clinical need? If this is not possible/relevant, then I wondered about severity of disease. There was some information provided on the types of eye conditions, but as a non-clinician I wasn’t certain about the severity/prognosis. Is it possible to group these conditions into relevant severity groups and consider that as a possible confounder and/or effect modifier in the analysis? Clinical need was discussed in the discussion section, but I think this could be further expanded, and perhaps sensitivity analyses around some of these other variables that could indicate clinical need could be explored.

I know families that had not received care in the 6 months prior to the portal launch were excluded, but is there any information on how long the patients have been registered/attending the specialist centre and might that possibly be a predictor of portal use? I also wondered about inpatient vs outpatient attendance, were these types of appointments treated the same? 

Many of the variables were ‘binarized’, but could the rationale for this be described further? Could collapsed categories have been considered (e.g. for ethnicity, using a version of the 5+1 census categories?) This was referred to in the discussion section on limitations but could be expanded. 

Results 

Table 1 – would be helpful if all variables included in the logistic models could be reported here (e.g. non-attendance to hospital appointments, etc). I also thought Table S1 was more informative than table 1 – the authors may wish to consider switching these, and perhaps including the chisq/p-values for the bivariate relationships.

Missing data was reported overall – but what about missing data by variable? Were there any variables in particular with a large proportion of missing data? Are the assumptions of complete-case analysis likely met and/or were any sensitivity analyses considered?

Table 2 and 3 – it looks like some of the ethnicity categories are missing, could these be included?

Line 141 – might there be a typo in the 95% CI or the p-value for the adjusted OR for the ‘Black Other’ ethnicity category? 

Discussion

The discussion covered many relevant factors to consider, but it would be helpful to get a broader sense of the literature and how this work might fit in to other studies about activation/uptake of patient portals.

Useful recommendations (e.g. incorporating non-digital interventions) were provided in the conclusions, but these could be discussed in more detail in the main discussion section.

Minor comments

- There is a period missing from the “author summary” paragraph, line 61.

- Table 2, ‘Asian Indian’ has 3 SD recorded for the 95% CI, all others have 2 SD.

Many thanks for the opportunity to review this paper and I do hope these comments are helpful to the authors.

Reviewer #2: Thank you for the opportunity to review the manuscript, "Socioeconomic and demographic patterning of family uptake of a paediatric electronic patient portal innovation", submitted to PLOS Digital Health as a research paper by Dr. Solebo and colleagues. The paper presents the findings from a single site study of use among children and their families of a novel patient portal at a single site. The paper is timely, relevant, detailed, and well-written. The manuscript is also well organized and follows the journal guidelines. There are some items the authors should consider modifying. These follow.

Foremost, the study is framed by an introduction and the discussion section focused on a basic exposition to the influence of health information technology at the international level, including barriers to access in low resource countries. This discussion is interesting and reasonably well referenced, but unrelated to the study itself. Given the study focus on patient portal use and personal health records access among children and their families, the introduction should focus on that topic, presenting concisely, what is known and where the gaps are that this particular study will address. The current introduction does not clearly indicate the motivation for the presented study.

The manuscript presents a clear justification for the use of a population of children under the age of 18 seen in an optimal logic clinic, however, the larger manuscript repeatedly refers to the population as involving children with complex medical conditions. This is confusing to the reader. While this reviewer respects the idea that Complex eye disease has value as a burden on patients and their families, most readers are likely to imagine the concept of complex medical conditions in children referring to illnesses such as cancer, hematologic diseases, or diabetes that involves multiple organs. The manuscript should be precise about its focus on eye disease from the outset and throughout. With that said, and in this reviewer's experience, complex multi organ chronic disease among children tends to be a stronger motivator for patient portal use than single organ disease. The authors should also ensure that their literature review captures the specific state of knowledge as it relates to pediatric patient portal use among children with chronic eye disease, which currently is not presented in the introduction section.

The manuscript presents findings from different childhood age groups but does not justify how the age groups are combined. Further, there is no information about children aged 16 to 18 despite the methods indicating that children with this age group will be included in the study.

The manuscript indicates that they developed a home grown patient portal called MyGOSH. The manuscript does not describe the portal, or indicate why they chose a homegrown portal – which may limit generalizability – over the use of the integrated my chart portal that is available for their epic EHR system.

Reviewer #3: This was a generally well-written article. With correct statistical analysis, and a clear message that ensure equitable digital health platforms to minimis inequalities in healthcare utilisation. 

The sequencing of the sections may need to fit in better with the PLOSOne format. 

From line 292-376 Material and methods, Study design, Inclusion/exclusion criteria, Data collection, Statistical analyses should be before the Results and Discussion section. 

Minor considerations

Referencing 

In general, the paper may be improved with more appropriate references: 

• Line 227-228 – how was non-engagement deduced as a ‘soft marker’? 

• Line 260-261 – are there any peer-reviewed metrics that the authors could consider utilising to understand the perceived need for using the portal? If so please provide the reference. If not, could authors consider qualitative or consumer designed approaches. 

• The authors may benefit from including some reference around the need for consumer engagement in co-designing digital health intervention and portals. 

The following references may be helpful: 

Greenhalgh T, Hinton L, Finlay T, Macfarlane A, Fahy N, Clyde B, et al. Frameworks for supporting patient and public involvement in research: Systematic review and co-design pilot. Health Expectations. 2019;22(4):785-801.

Otherwise, great article!

6. PLOS authors have the option to publish the peer review history of their article (what does this mean?). If published, this will include your full peer review and any attached files.

**Do you want your identity to be public for this peer review?** For information about this choice, including consent withdrawal, please see our Privacy Policy.

Reviewer #1: No

Reviewer #2: No

Reviewer #3: No

---

## [Decision Letter · Decision Letter 1]

7 Aug 2024

PDIG-D-24-00122R1

Socioeconomic and demographic patterning of family uptake of a paediatric electronic patient portal innovation

PLOS Digital Health

Dear Dr. Solebo,

Thank you for submitting your manuscript to PLOS Digital Health. After careful consideration, we feel that it has merit but does not fully meet PLOS Digital Health's publication criteria as it currently stands. Therefore, we invite you to submit a revised version of the manuscript that addresses the points raised during the review process.

Please submit your revised manuscript within 30 days Sep 06 2024 11:59PM. If you will need more time than this to complete your revisions, please reply to this message or contact the journal office at digitalhealth@plos.org. Please include the following items when submitting your revised manuscript:

We look forward to receiving your revised manuscript.

Kind regards,

Calvin Or, PhD

Section Editor

PLOS Digital Health

Journal Requirements:

Additional Editor Comments (if provided):

Reviewers' comments:

Reviewer's Responses to Questions

**Comments to the Author**

1. If the authors have adequately addressed your comments raised in a previous round of review and you feel that this manuscript is now acceptable for publication, you may indicate that here to bypass the “Comments to the Author” section, enter your conflict of interest statement in the “Confidential to Editor” section, and submit your "Accept" recommendation.

Reviewer #1: All comments have been addressed

Reviewer #2: All comments have been addressed

2. Does this manuscript meet PLOS Digital Health’s publication criteria? Is the manuscript technically sound, and do the data support the conclusions? The manuscript must describe methodologically and ethically rigorous research with conclusions that are appropriately drawn based on the data presented.

Reviewer #1: Yes

Reviewer #2: Yes

3. Has the statistical analysis been performed appropriately and rigorously?

Reviewer #1: Yes

Reviewer #2: Yes

4. Have the authors made all data underlying the findings in their manuscript fully available (please refer to the Data Availability Statement at the start of the manuscript PDF file)?

Reviewer #1: Yes

Reviewer #2: No

5. Is the manuscript presented in an intelligible fashion and written in standard English?

Reviewer #1: Yes

Reviewer #2: Yes

6. Review Comments to the Author

Reviewer #1: Many thanks for your attention to the previous comments made - not just mine, but the other reviewers as well. I believe this manuscript has been greatly improved by the revisions and have no additional comments or suggestions.

Reviewer #2: The authors have been extremely responsive to prior review and the manuscript is much tighter. The only remaining item the authors should consider is that it remains confusing that there is an apparent focus on the single chronic medical condition of blindness, while the study itself appears to include all comers (i.e., not just patients with blindness). This is confusing to the reader.

7. PLOS authors have the option to publish the peer review history of their article (what does this mean?). If published, this will include your full peer review and any attached files.

**Do you want your identity to be public for this peer review?** For information about this choice, including consent withdrawal, please see our Privacy Policy. 

Reviewer #1: No

Reviewer #2: Yes: S. Trent Rosenbloom

---

## [Editor Report · Decision Letter 2]

23 Aug 2024

Socioeconomic and demographic patterning of family uptake of a paediatric electronic patient portal innovation

PDIG-D-24-00122R2

Dear Dr Solebo,

We are pleased to inform you that your manuscript 'Socioeconomic and demographic patterning of family uptake of a paediatric electronic patient portal innovation' has been provisionally accepted for publication in PLOS Digital Health.

Best regards,

Calvin Or, PhD

Section Editor

PLOS Digital Health